# The Mitogen-Activated Protein Kinase PlMAPK2 Is Involved in Zoosporogenesis and Pathogenicity of *Peronophythora*
*litchii*

**DOI:** 10.3390/ijms22073524

**Published:** 2021-03-29

**Authors:** Jiamin Huang, Pinggen Xi, Yizhen Deng, Weixiong Huang, Jingrui Wang, Qingqing Zhao, Wensheng Yang, Wen Li, Junjian Situ, Liqun Jiang, Tianfang Guan, Minhui Li, Zide Jiang, Guanghui Kong

**Affiliations:** Guangdong Province Key Laboratory of Microbial Signals and Disease Control, Department of Plant Pathology, South China Agricultural University, Guangzhou 510642, China; yeah_159@163.com (J.H.); xpg@scau.edu.cn (P.X.); dengyz@scau.edu.cn (Y.D.); qq184863352@163.com (W.H.); wangjr202103@163.com (J.W.); zqq26297185@163.com (Q.Z.); ywshylove@gmail.com (W.Y.); SLLDLZ@163.com (W.L.); junjian.st@hotmail.com (J.S.); jiangliqun@gdaas.cn (L.J.); guantianfang@163.com (T.G.); liminhui@scau.edu.cn (M.L.); zdjiang@scau.edu.cn (Z.J.)

**Keywords:** MAPK, *Peronophythora litchii*, zoosporogenesis, sporangial cleavage, pathogenicity, CRISPR/Cas9

## Abstract

As an evolutionarily conserved pathway, mitogen-activated protein kinase (MAPK) cascades function as the key signal transducers that convey information by protein phosphorylation. Here we identified *PlMAPK2* as one of 14 predicted MAPKs encoding genes in the plant pathogenic oomycete *Peronophythora litchii*. *PlMAPK2* is conserved in *P.*
*litchii* and *Phytophthora* species. We found that *PlMAPK2* was up-regulated in sporangium, zoospore, cyst, cyst germination and early stage of infection. We generated *PlMAPK2* knockout mutants using the CRISPR/Cas9 method. Compared with wild-type strain, the *PlMAPK2* mutants showed no significant difference in vegetative growth, oospore production and sensitivity to various abiotic stresses. However, the sporangium release was severely impaired. We further found that the cleavage of the cytoplasm into uninucleate zoospores was disrupted in the *PlMAPK2* mutants, and this developmental phenotype was accompanied by reduction in the transcription levels of *PlMAD1* and *PlMYB1* genes. Meanwhile, the *PlMAPK2* mutants exhibited lower laccase activity and reduced virulence to lychee leaves. Overall, this study identified a MAPK that is critical for zoosporogenesis by regulating the sporangial cleavage and pathogenicity of *P.*
*litchii*, likely by regulating laccase activity.

## 1. Introduction

Oomycetes are fungus-like eukaryotic microorganisms which are evolutionarily close to photosynthetic algae, but are distant from fungi [1,2]. Oomycete contains a large number of economically significant pathogens in agriculture, forestry and the ecosystem. For example, *Phytophthora infestans*, *Phytophthora capsici*, *Phytophthora sojae* and *Peronophythora litchii* severely damage potato, soybean, cucurbits and lychee, respectively [3,4]. Among them, lychee downy blight caused by *P.*
*litchii* is the most destructive disease of lychee and results in crop losses ranging from 30% to 80% annually [5].

As evolutionarily conserved signaling pathways, mitogen-activated protein kinase (MAPK) cascades function as key signal transducers in plants, animals, fungi and oomycetes, which utilize protein phosphorylation/dephosphorylation cycles to channel information [6]. In all eukaryotes, MAPK cascades generally consist of three interlinked protein kinases (PKs) [7]. Activated MAPK kinases (MAP3Ks) first phosphorylate two Ser and/or Thr residues of MAPK kinases (MAP2Ks). Activated MAP2Ks trigger MAPK activation at the well-conserved threonine-x-tyrosine (TXY) motif [6,8]. Activated MAPKs can phosphorylate downstream substrates, regulating their functions [6].

MAPK cascades participate in the response to diverse stresses, developmental processes and infection of fungi and oomycetes [9,10]. In *P. sojae*, PsMAPK1 is involved in vegetative growth, zoosporogenesis and virulence [11]. Silencing of *PsSAK1* led to defects in zoospore viability and infection of soybean [12]. PsMPK7 is required for stress tolerance, detoxification of reactive oxygen species (ROS), cyst germination, sexual reproduction and pathogenicity [13]. Silencing of *P.*
*litchii MAPK10* led to defects in mycelial growth, sporulation, laccase activity and plant infection [10]. These studies demonstrated that MAPK signaling is critical for development and pathogenicity of plant pathogenic oomycete; however, the MAPK pathways in oomycetes remain largely unknown.

In this study, we identified the *P.*
*litchii MAPK2* gene, a homologue of *Saccharomyces cerevisiae Fus3*, conserved in *P.*
*litchii* and *Phytophthora* species. *PlMAPK2* is up-regulated in asexual stages and early stage of infection. Knockout of *PlMAPK2* impaired the zoospore release, due to failure of the sporangial cleavage. *PlMAPK2* is also involved in the pathogenicity and laccase activity of *P.*
*litchii*.

## 2. Results

### 2.1. PlMAPK2 Is Conserved in P. litchii and Phytophthora Species and Up-Regulated in Asexual Development and Early Stage of Infection

Bioinformatic analysis showed that *P. litchii* encodes 14 MAPKs (Appendix A). Here we characterized the functions of *P.*
*litchii* mitogen-activated protein kinase, PlMAPK2. We searched the PlMAPK2 orthologs in 13 *Phytophthora* species and found PlMAPK2 orthologs in all tested species (with 91–96% similarities), indicating that PlMAPK2 orthologs are highly conserved in *Phytophthora* species (Figure 1A and Appendix A; Appendix A); however, their function has not been reported. PlMAPK2 is a MAPK with the conserved TEY motif within the catalytic domain of the serine/threonine kinase located in 88-415 amino acid in the full length of 455 amino acids (Figure 1A). A phylogenetic tree was constructed based on the protein sequences of PlMAPK2 and the orthologs from oomycetes and fungi, showing that PlMAPK2 likely belong to the Kss1/Fus3-type MAPK with *Saccharomyces cerevisiae* Fus3 (Appendix A).

Transcriptional analysis found that *PlMAPK2* is up-regulated in sporangium (SP), cyst (CY), germination of cyst (GC) and early stage of infection (Figure 1B). These results suggest that PlMAPK2 might function in asexual development and infection stages.

### 2.2. Generation of PlMAPK2 Knockout Mutants Using CRISPR/Cas9 Genome Editing Technology

To investigate function of PlMAPK2, we generated *PlMAPK2* knockout mutants using the CRISPR/Cas9 system. Single-guide RNAs were designed to disrupt the *PlMAPK2* coding region (Figure 2A). The transformants were screened by G418 resistance and then verified by genomic PCR and sequencing (Figure 2B,C). Finally, two *PlMAPK2* mutants (M113, M115) were obtained, and a transformant that failed to acquire *PlMAPK2* mutation was selected as the control (CK). We also confirmed the transcription of *PlMAPK2* in WT (wild-type stain, SHS3) and *PlMAPK2* mutants, and our results showed that *PlMAPK2* did not express in M113 and M115 (Figure 2D). These results demonstrated that *PlMAPK2* was knocked out in M113 and M115.

### 2.3. Knockout of PlMAPK2 Affects Zoosporogenesis of P. litchii

At the asexual life cycle stages, oomycetes produce sporangia, which then release zoospores. To investigate the function of PlMAPK2 at the asexual development, the *PlMAPK2* mutants, CK and WT, were cultured in CJA medium for 5 days, and then the sporangia were collected and calculated. We found that the knockout of *PlMAPK2* did not affect the production of sporangia (Figure 3A,B). The sporangia produced by *PlMAPK2* mutants showed similar lengths and widths compared with WT (Figure 3C,D). Zoospore release rates were observed 0.5 and 2 h after collecting the sporangia in water. Around 87% of WT sporangia released zoospores. However, only about 2% of M113 or M115 sporangia released zoospores, in 2 h (Figure 3A,E). Therefore, our results indicated that PlMAPK2 affected the zoospores release, but not the production of sporangia.

### 2.4. PlMAPK2 Is Required for Sporangial Cleavage during Zoospore Development

Zoospore development involves cleavage of the sporangial cytoplasm by nucleus-enveloping membrane networks [14], we next assessed sporangial cleavage in WT and *PlMAPK2* mutants by FM4-64 and DAPI staining. We found that the sporangial cleavage was impaired in *PlMAPK2* mutants. The nuclei within the WT sporangia were regularly spaced and the cytoplasm differentiated to form developed zoospores. In contrast, the sporangium cytoplasm of the *PlMAPK2* mutants remained undifferentiated and nuclei remained disordered (Figure 4), so there were few developed zoospores formed and released in *PlMAPK2* mutants. Our results demonstrated that deletion of *PlMAPK2* impaired the sporangial cleavage during zoospore development, leading to defects of zoospore release.

### 2.5. PlMAD1 and PlMYB1 Were Down-Regulated in PlMAPK2 Mutants

*P. sojae* is phylogenetically close to *P.*
*litchii*. In *P.*
*sojae*, a MADS-box transcription factor PsMAD1 and a Myb transcription factor PsMYB1 were reported to be involved in cleavage of sporangium and virulence [15,16]. We identified the *P. litchii* orthologs of MAD1 and MYB1, named as PlMAD1 and PlMYB1, respectively (Appendix A). These proteins share sequence identities of 80 and 82% respectively with their counterparts in *P.*
*sojae*. Here, the total RNA was extracted from sporangia of WT and *PlMAPK2* mutants. Then, we evaluated the transcriptional levels of those two genes at the stage of sporangia by qRT-PCR. Our results showed that *PlMAD1* and *PlMYB1* were down-regulated in *PlMAPK2* mutants (Figure 5), suggesting that PlMAPK2 affects transcriptional levels of *PlMAD1* and *PlMYB1*.

### 2.6. PlMAPK2 Is Required for Full Virulence and Regulates Laccase Activity in P. litchii

*PlMAPK2* is up-regulated in the early stage of infection; therefore, PlMAPK2 might be critical for infection. The virulence of WT and *PlMAPK2* mutants were tested on lychee leaves. Our results showed that the lesion length of *PlMAPK2* mutants is reduced by 33–54% compared with WT (Figure 6). These results indicate that PlMAPK2 is required for the full virulence of *P.*
*litchii*.

In fungi, laccases participate in the oxidation of antibiotics such as flavonoids and phytoalexins and contribute to the virulence of pathogen [17]. In oomycetes, the laccase activity is also associated with the plant infection [5,10,18]. Here we tested the laccase activity of *PlMAPK2* mutants and found that these mutants showed decreased laccase activity (Figure 7A). Therefore, we conclude that PlMAPK2 is critical for the virulence and regulates laccase activity in *P.*
*litchii*.

We further examined the transcriptional levels of 8 laccase-encoding genes, and found that 3 of them, *Pl104952*, *Pl106183* and *Pl111417*, were dramatically down-regulated in *PlMAPK2* mutants, while *Pl103272* and *Pl106923* were slightly up-regulated in *PlMAPK2* mutants (Figure 7B); the transcription level of Pl106181, Pl 106924 and Pl111416 showed no significant difference in *PlMAPK2* mutant. We inferred that PlMAPK2 affected the transcription levels of these laccase-encoding genes.

### 2.7. Assessment of Growth and Stress Response in the PlMAPK2 Mutants

PlMAPK2 showed 53% similarity with Fus3 of *S.*
*cerevisiae*. In *S.*
*cerevisiae*, Fus3 involved in filamentous growth and mating [6]. Here, we tested the growth, production of sexual spores (oospores) and responses to various stresses (CFW, SDS, CR, H_2_O_2_, NaCl and CaCl_2_). Compared to the WT, the mutants showed no significant difference of growth rate on CJA medium. The *PlMAPK2* mutants were exposed to various stresses, and we found that *PlMAPK2* mutants showed no significance in responses to various stresses comparing with WT (Figure 8). We further found that the oospore size and production of the *PlMAPK2* mutants were also similar to that of WT (Figure 8 and Appendix A). These results suggested that deletion of *PlMAPK2* did not affect the filamentous growth, responses to various stresses, or sexual reproduction in *P.*
*litchii*.

## 3. Discussion

MAPK cascades are widespread and conserved in plants, fungi and oomycetes. However, the function of MAPKs in development and pathogenicity of oomycetes are rarely known. In oomycetes, PsMAPK1, PsSAK1, PsMPK7 and MAPK10 showed various functions in vegetative growth, zoospore viability, stress tolerance, detoxification of reactive oxygen species, cyst germination, sexual reproduction, laccase activity and pathogenicity [10,11,12,13]. In this study, we identified the PlMAPK2 and found that its orthologs in oomycete are functional unknown MAPKs. We found that PlMAPK2 plays a key role in zoosporogenesis by regulating the sporangial cleavage, PlMAPK2 also contributes to the pathogenicity and is associated with laccase activity of this pathogen. This study firstly reported a MAPK required for sporangia cleavage in oomycete, providing new insight into the functions of MAPK signaling pathway in oomycete life cycle.

Zoospore release is important for plant pathogenic oomycete life cycle, which is associated with the sporangial cytoplasm cleavage and the assembly of flagella [14]. CDC14 orthologs have been revealed to function in sporulation in pathogenic oomycetes *P.*
*infestans* and *P.*
*sojae* [19,20]. MAD-box and MYB transcription factors were also shown to be essential for sporangial cleavage in *P.*
*sojae* [15,16]. Here, our results showed that *PlMAPK2* mutation led to defects in sporangial cleavage and down-regulation of PlMAD1 and PlMYB1. Therefore, we hypothesize that PlMAPK2 regulates zoospore release through transcription factors including MAD1 and MYB1. However, further studies are needed to reveal the zoospore formation process.

In *S.*
*cerevisiae*, MAPK kinase cascades always directly regulate transcription factors via phosphorylation [6], therefore indirectly modulating the transcription of down-stream genes. In oomycetes, silencing of a *P.*
*sojae MAPK-encoding* gene, *PsSAK1*, decreased the transcriptional level of *PsMYB1*. Here, we found that the expression of *PlMAD1* and *PlMYB1* was down-regulated in *PlMAK2* mutants (Figure 5), which suggested that PlMAPK2 might regulate zoosporogenesis and virulence by regulating *PlMAD1* and *PlMYB1* at expression level. Additional experiments are needed to clarify how MAPK signaling regulates sporangial development and the activation of pathogenic pathways in oomycetes.

Knockout of *PlMAPK2* decreased the pathogenicity, the laccase activity and interfered the transcription of several laccase-encoding genes in *P.*
*litchii*. These results are consistent with previous reports that laccase activity is important for plant pathogenic fungi and oomycetes [10,21]. Further, we analyzed the expressions of laccase-encoding genes; three of them, *Pl104952*, *Pl106183* and *Pl111417*, are significantly down-regulated in the *PlMAPK2* mutants (Figure 7B). We speculate that PlMAPK2 might regulate laccase activity by affecting the transcription of laccase-encoding genes in *P.*
*litchii*. PlMAPK10 also regulated the transcription level of another laccase-encoding gene [10]. These results suggested that different MAPKs might specifically regulate corresponding laccase-encoding genes.

In *S.*
*cerevisiae*, Fus3 was involved in mating and filamentous growth [6], and in *Magnaporthe oryzae*, Pmk1, the homologue of Fus3, was involved in appressorium formation and penetration and invasive growth [8]. In *P.*
*litchii*, PlMAPK2 showed highest similarity with Fus3 of *S.*
*cerevisiae*, but PlMAPK2 was not associated with growth, sexual development and responses to various tolerance stresses of *P.*
*litchii* (Figure 8). We found that PlMAPK2 is important for the zoospore release and virulence of this pathogen (Figure 3). These results indicated that sequence and functional differentiation occurred in fungi and oomycete MAPK cascades for their specific development and pathogenic processes.

## 4. Materials and Methods

### 4.1. Sequences and Phylogenetic Analysis

Genome sequence of *P.*
*litchii* was obtained from NCBI (BioProject ID: PRJNA290406) [22]. Genome sequence of *Pythium* species (Pythium vexans DAOM BR484, Pythium aphanidermatum DAOM BR444, Pythium irregulare DAOM BR486) were obtained from the *Pythium* genome database (http://protists.ensembl.org/index.html, accessed on 6 July 2020). Other oomycete and fungal sequences were obtained from NCBI (http://www.ncbi.nlm.nih.gov, accessed on 9 June 2020). The obtained protein sequences were submitted to NCBI-CDD (http://www.ncbi.nlm.nih.gov/Structure/cdd, accessed on 9 July 2020) to identify conserved domains. All sequences used in this study were listed in Appendix A.

Sequence alignments were created using ClustalW program [23] and the phylogenetic tree was constructed with MEGA 6.0 program using a neighbor-joining algorithm with 1000 bootstrap replicates [24].

### 4.2. Strains and Culture Conditions

*Peronophythora litchii* strain SHS3 were locally maintained on CJA medium (juice from 200 g carrot topped up to 1 L, 15 g/L agar for solid media) [25] at 25 °C in the dark.

### 4.3. Transcriptional Analysis

Total RNA was extracted using the total RNA kit (catalog [cat.] number R6834-01; Omega). Samples included mycelia (MY), sporangia (SP), zoospores (ZO), cysts (CY), germination of cysts (GC) and 1.5–24 h post inoculation. The first-strand cDNA was synthesized from total RNA by oligo(dT) priming using a Moloney murine leukemia virus (MMLV) reverse transcriptase kit (number S28025-014; Invitrogen, Carlsbad, CA, USA). Transcription of *PlMAPK2* was analyzed with qRT-PCR (Quantitative Reverse transcription PCR) assay using primers MAPK2-qRTF and MAPK2-qRTR on ABI Prism 7500 Fast real-time PCR System (Applied Biosystems, Foster City, CA, USA). *P.*
*litchii* actin gene was used as loading control and the relative fold change was calculated using the 2^−^^∆∆*C*t^ method [26]. Transcription of *PlMAD1* (*P.*
*litchii* MAD-box transcription factor 1), *PlMYB1* (*P.*
*litchii* MYB transcription factor 1) and laccase-encoding genes were also analyzed as described above, primers were listed in Appendix A.

### 4.4. CRISPR/Cas9 Editing for PlMAPK2

To generate *PlMAPK2* knockout transformants, *P.*
*litchii* protoplasts were transformed using the polyethylene glycol (PEG)–CaCl_2_-mediated method as described previously [25,27,28]. The plasmids, pYF2-PsNLS-hSpCas9, pYF2.3G-RibosgRNA::*PlMAPK2* and pBluescript II KS(+)::*PlMAPK2* were cotransformed into protoplasts of *P.*
*litchii* (Figure 1A and Appendix A). We designed two sgRNAs (sgRNA1 and sgRNA2) that target *PlMAPK2* using the protocol described previously [27], then checking their RNA secondary structure online (http://rna.urmc.rochester.edu/RNAstructureWeb/Servers/Predict1/Predict1.html, accessed on 9 July 2020). The transformed protoplasts were regenerated overnight, and the recovered mycelia were selected on CJA medium with 30 μg/mL G418. After 2–3 days, the primary transformants were transferred to new selective medium and identified by subsequent genomic PCR (Polymerase chain reaction) and sequencing analyses. The primers used to genomic PCR are listed in Appendix A.

### 4.5. Analysis of Sporangium, Zoospore and Oospore Development

Sporangia were harvested by flooding the mycelia, which had been cultured on CJA media for 5 days, with sterile water, and then filtering the suspension through a 100 μm strainer [25]. The suspension was incubated with sterile distilled water at 16 °C for 2 h to release zoospores. For oospore observation, *P.*
*litchii* strains were inoculated on CJA medium for 10 days, then the production and diameter of oospores were calculated.

### 4.6. Chemical Dyes with FM4-64 and DAPI (4’,6-Diamidino-2-Phenylindole, Dilactate)

Sporangium were visualized by the blue-fluorescent nucleic acid stain 4′, 6-diamidino-2-phenylindole (DAPI), dilactate (D3571, Invitrogen, USA) and then viewed with Olympus BX53F microscope as described previously [15]. Red-fluorescent FM 4-64 dye (T13320, Invitrogen, USA) was used to analyze the zoospore membrane in sporangium [16].

### 4.7. Pathogenicity Test

Five-mm mycelium plugs of the *PlMAPK2* mutants were inoculated on the abaxial surface of lychee leaves in a dark box and then were placed in a climate room under 80% humidity in the dark at 25 °C; lesion lengths were measured after 48 h. Inoculation with sporangia: the sporangia were collected as described above and we adjusted the concentration of sporangia to 10 sporangia per μL, then 10 μL of sporangium suspension were inoculated on abaxial surface of the leaves, the lesion lengths were measured after 48 h. WT and CK strains were used as controls.

### 4.8. Laccase Activity Assays

The laccase activity was analyzed following the previously described procedure [18]. The experiments were repeated three times independently, with three technical replicates each time.

## 5. Conclusions

In summary, we characterized the functions of PlMAPK2 in *P.*
*litchii* with CRISPR/Cas9-mediated genome editing method. PlMAPK2 plays a key role in zoosporogenesis, virulence and regulation of laccase activity and transcription of *PlMAD1*, *PlMYB1* in *P.*
*litchii*. This study provided new insight into the functions of MAPK signaling in zoosporogenesis and pathogenicity of oomycete, and identified MAPK2 as a potential fungicide target for oomycete pathogen control.

## Figures and Tables

**Figure 1 ijms-22-03524-f001:**
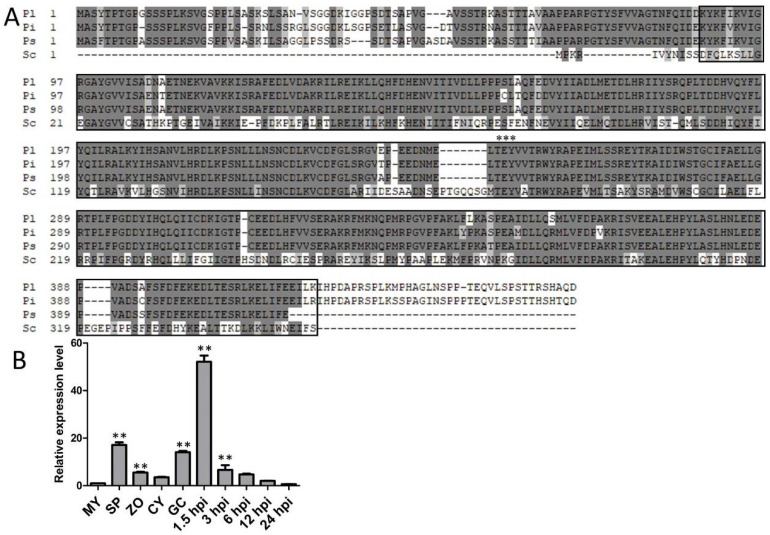
Identification of PlMAPK2 in *P. litchii*. (**A**) Amino acid sequences alignment of *PlMAPK2* (Pl) and its orthologs from *P.*
*infestans* (Pi), *P.*
*sojae* (Ps) and *S.*
*cerevisiae* (Sc). The black boxes and three black asterisks represent catalytic domain of the serine/threonine kinase and conserved TEY motif, respectively. (**B**) Expression pattern of *PlMAPK2* during the asexual life cycle and infection stages was analyzed by quantitative reverse transcription PCR (qRT-PCR). MY: Mycelia; SP: sporangia; ZO: zoospore; CY: cyst; GC: germination of cyst; hpi: hours post inoculation. Relative expression levels were calculated using the MY values as reference. Asterisks indicate significant difference (** *p* < 0.01) based on Bonferroni’s multiple comparison test. These experiments were repeated three times.

**Figure 2 ijms-22-03524-f002:**
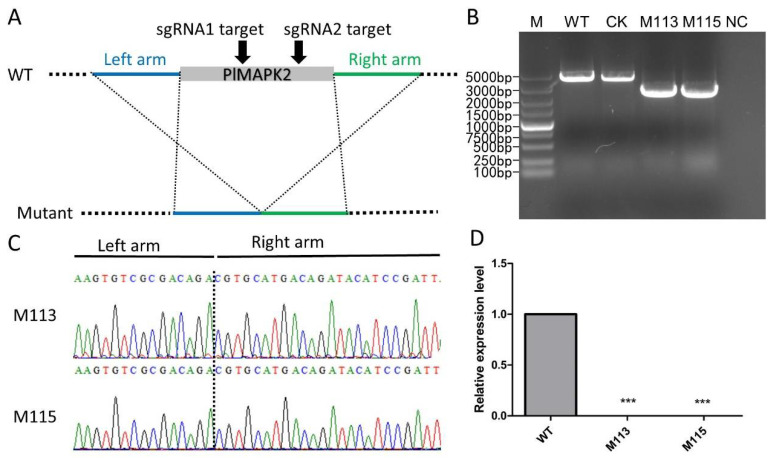
CRISPR/Cas9-mediated deletion of *PlMAPK2* in *P.*
*litchii*. (**A**) Schematic representation of the strategy of CRISPR/Cas9-mediated mutagenesis of *PlMAPK2*. Two single-guide RNAs targeted the *PlMAPK2* gene sequence (indicated by black arrows). The *PlMAPK2* mutants were identified by genomic PCR (**B**) and sequencing (**C**). (**D**) *PlMAPK2* was not expressed in *PlMAPK2* mutants. WT: wild type; CK: the transformant failed to acquire *PIMAPK2* mutation; NC: negative control (distilled water as template). “***” indicates significant difference (*t*-test, *p* < 0.001). This experiment was repeated three times.

**Figure 3 ijms-22-03524-f003:**
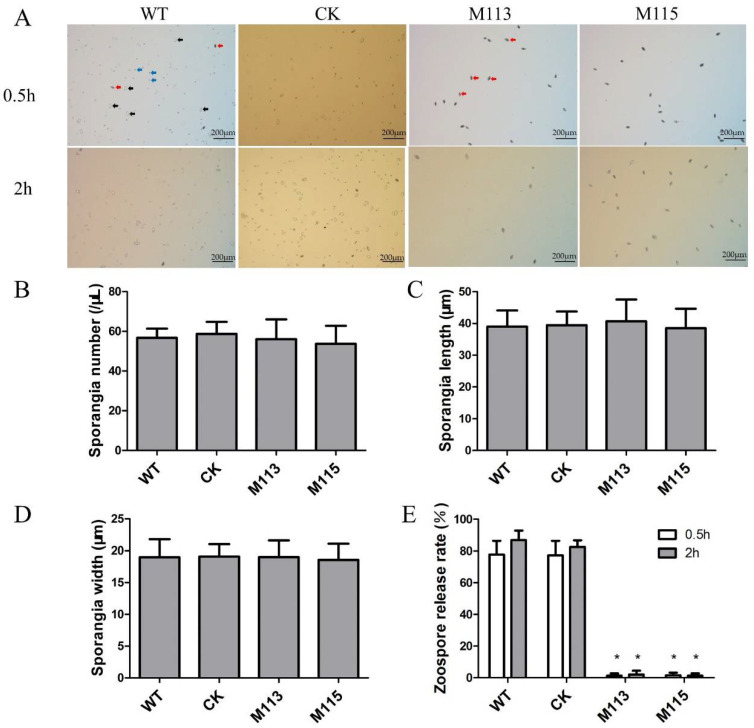
Knockout of *PlMAPK2* decreased the zoospores release rate. *PlMAPK2* mutants, CK (the transformant failed to acquire *PIMAPK2* mutation) and WT (wild type) were cultured on CJA medium for 7 days and sporangia were collected and used for releasing zoospores. (**A**) Photographs were taken 0.5 and 2 h after releasing zoospores. Scale bar = 200 μm. Black arrows indicate the representative released sporangia; red arrows indicate representative unreleased sporangia; blue arrows indicate representative zoospores. (**B**) The sporangia numbers were calculated. (**C**,**D**) The sporangia lengths and widths were measured. (**E**) The zoospore release rates of *PlMAPK2* mutants, CK and WT were calculated after 0.5 and 2 h. Asterisks indicate significant difference between WT and *PlMAPK2* mutants (*t*-test, *p* < 0.01). These experiments were replicated three times.

**Figure 4 ijms-22-03524-f004:**
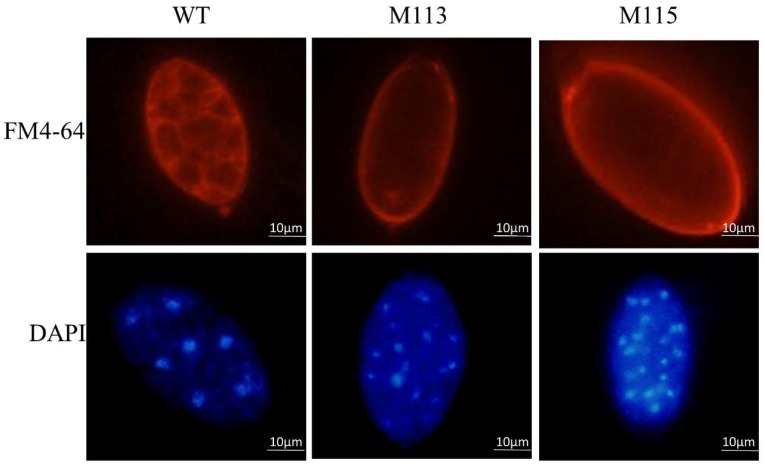
Assessment of nuclei distribution and cytoplasm cleavage within sporangia. The nuclei in sporangia were detected by DAPI staining. Plasma membrane stained by FM4-64. Representative microscopic images for WT (wild type) or the mutants were displayed (*n* ≥ 9). Scale bar = 10 μm.

**Figure 5 ijms-22-03524-f005:**
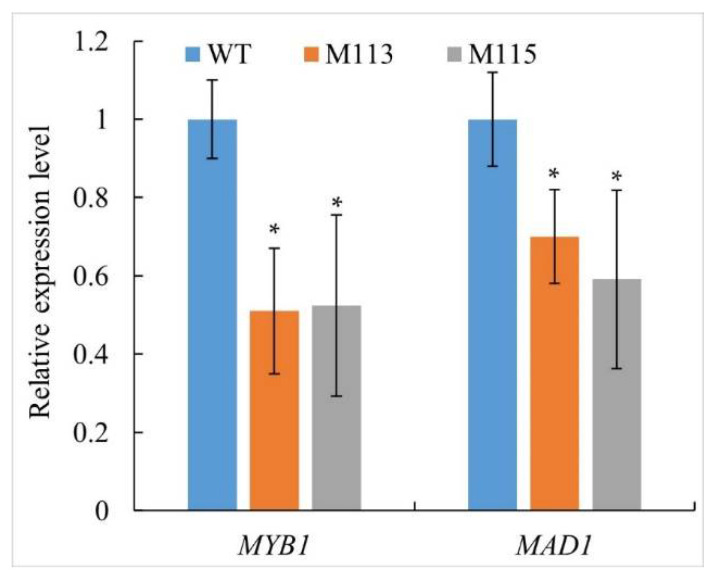
*PlMAD1* and *PlMYB1* is down-regulated in sporangia of *PlMAPK2* mutants, the expression of *PlMAD1* and *PlMYB1* were analyzed by qRT-PCR. WT (wild type) was used as control. Asterisks indicate significant difference (*t*-test, *p* < 0.01). These experiments were replicated three times.

**Figure 6 ijms-22-03524-f006:**
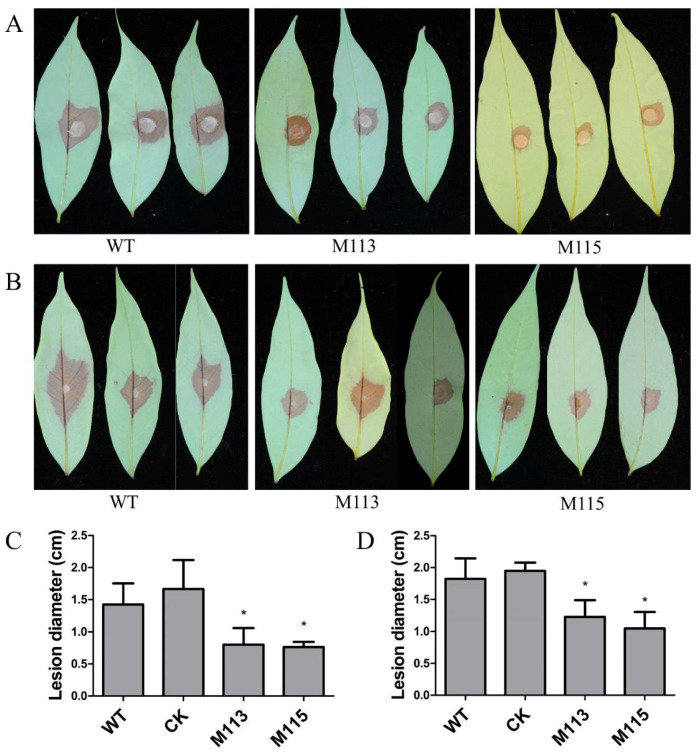
Infection assays with WT and the *PlMAPK2* mutants. The virulence of *PlMAPK2* mutants were tested on lychee leaves, with WT (wild type) as the positive control. Mycelium plugs (**A**) and sporangia (**B**) were inoculated on the lychee leaves, respectively. Photographs were taken 48 h after inoculation. (**C**,**D**) The lesion lengths were calculated 2 days after inoculation, corresponding to (**A**,**B**), respectively. CK: the transformant failed to acquire *PIMAPK2* mutation. Asterisks indicate significant difference (*t*-test, *p* < 0.01). These experiments were replicated three times, 5 leaves were calculated for each strain each time.

**Figure 7 ijms-22-03524-f007:**
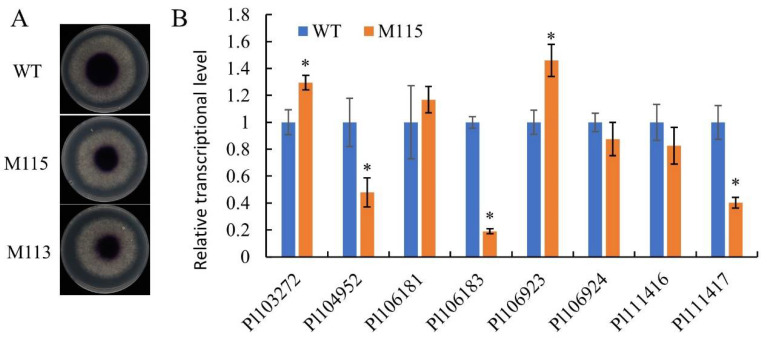
PlMAPK2 regulates laccase activity. (**A**) Mycelial mats of the aforementioned strains were inoculated on media containing 0.2 mM ABTS for 10 days. (**B**) The expression of 8 laccase-encoding genes were analyzed by qRT-PCR. Asterisks indicate significant difference (*t*-test, *p* < 0.01). WT: wild type. The experiments contain three independent biological repeats, each of which contained three replicas.

**Figure 8 ijms-22-03524-f008:**
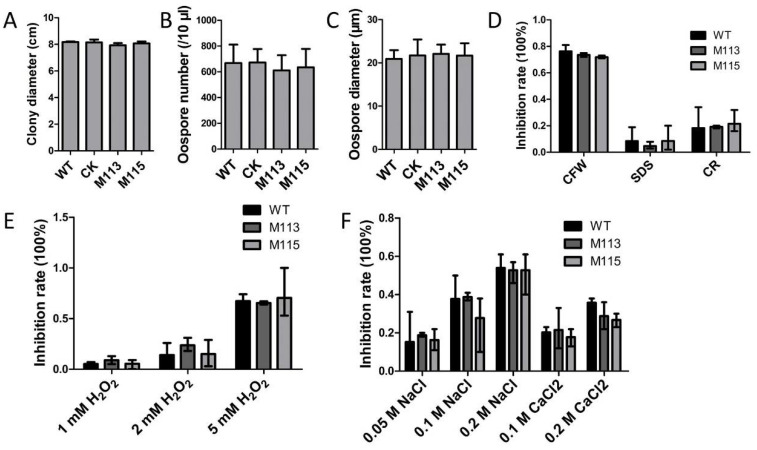
Assessment of mycelia growth, sexual reproduction and responses to various stresses. (**A**) Colony diameters of WT (wild type) or the *PlMAPK2* mutants were measured 5 days after inoculation on CJA medium. (**B**) Oospore number at 10 days after inoculation on CJA medium. (**C**) Oospores diameters of WT and *PlMAPK2* mutants at 10 days after inoculation on CJA medium. (**D**) Inhibition rates of WT and *PlMAPK2* mutants on Plich medium supplemented with 350 μg/mL CFW, 25 μg/mL SDS or 100 μg/mL CR. (**E**) Inhibition rates of WT and *PlMAPK2* mutants on Plich medium with 1, 2 and 5 mM H_2_O_2_. (**F**) Inhibition rates of WT and *PlMAPK2* mutants on Plich medium with 0.05 mM NaCl, 0.1 mM NaCl, 0.2 mM NaCl, 0.1 mM CaCl_2_ and 0.2 mM CaCl_2_. CK: the transformant failed to acquire *PIMAPK2* mutation. These experiments were repeated three times, with three replicas each time.

## Data Availability

Not applicable.

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
