# Peer review of "The Mitogen-Activated Protein Kinase PlMAPK2 Is Involved in Zoosporogenesis and Pathogenicity of Peronophythoralitchii"

_ijms, 2021, doi:10.3390/ijms22073524_

Round 1
Reviewer 1 Report
I'm glad to review the manuscript titled “The mitogen-activated protein kinase PlMAPK2 is involved in zoosporogenesis and pathogenicity of Peronophythora litchii”. In this study, the authors generated PlMAPK2 knockout mutants using CRIPSPR-cas9 method and investigated the role of PlMAPK2 in P. litchi developmental stages under in vitro growth and during infection. They found PlMAPK2 showed no significant impacts on its vegetative growth, oospore production, response to abiotic stresses but resulted in the defect of the cleavage of the cytoplasm into uninucleate zoospores and dramatically reduced rates of zoospore release. The PlMAPK2 knockout mutant also showed reduced virulence compared with the wild type strain. The authors also tried to investigate the impaired pathways that control the above phenomena. They identified two transcription factors that are regulated by PlMAPK2 and several genes coding enzymes with laccase activity.
The authors did a nice job telling a completed story of the role of PlMAPK2 in P. litchi life cycle and infection, which provides insights on the role of the conserved MAPK signaling transduction pathway in phytophthora species. However, some results can be improved, especially the molecular mechanisms that are regulated by the transcription factors responding to MAPK regulation. And some conclusions are not solid based on the evidence showed in the manuscript. In particular, there eight genes encode laccase in the P. litchi and showed different expression regulation, up-, down- and non-regulation in the PlMAPK2 knockout mutant. The authors only emphasized the three down-regulated laccase genes (Line 155) and the conclusion is very vague (Line 156 to 157).
I listed my major and minor comments line-by-line below. I hope my comments are helpful to improve the manuscript quality. I look forward to the revised manuscript.
First of all, please spell the full name of the WT (wild type) and CK (the transformant failed to acquire PIMAPK2 mutation) across the figure legend. Both were only mentioned and explained in Line 84-87. However, it is hard to find their meanings (i.e., CK) when just looking at individual figures.
Line 75: In Fig. 1B, I would run the statistical test to indicate the differential expression levels among different developmental stages and infection stages. Also, please describe the statistical methods used for significance calculation throughout the figures. Please note that P should be italic when it presents the significant levels.
L93: Please describe the “*” in the figure legend.
L108: I would show the scale bar on the figure as well. For Fig. 3e, I would indicate whether the significance is a pair-wise comparison between WT and mutant or not. Here, it’s easy to get confused by the current presentation method.
I’m curious that why sporangia numbers are similar between WT and mutant? I guess this might be explained by the process of the zoospore generation process. However, I didn’t find such a description in the manuscript. Furthermore, what are the large, dark dots showed in the mutant image? Such large, dark dots didn’t capture by the following comparison of sporangia numbers, lengths, and widths.
L111: I enjoyed reading this section, which clearly showed PlMAPK2 is required for sporangial cleavage during zoospore development. The images are beautiful. Again, I would label the scale bar in the image.
L 125: Are there only two transcription factors regulated by MAPK2? Is it possible there are other transcription factors that are also regulated by MAPK? Such predictions can be made by previous literature or RNA-seq data from other close species. And can be validated by similar experiments in this study. Furthermore, by inferring the regulating transcription factors and potential pathways, we authors should raise a hypothesis and explain why disrupting the MAPK2 results in the defect of zoospore release. The gene names of MYB1 and MAD1 should be italic in Fig. 5.
L136: Which tissues were used for qRT-PCR?
Line 145: How many samples are used for significance calculation in lesion comparison?
Line 151: The authors should provide more details on how laccase activity contributes to P. litchii plant infection.
Line 155: According to Fig. 7B, there are three laccase-encoding genes that are downregulated and two of them are upregulated. However, the authors only present part of the results, down-regulated laccase-encoding genes, in Fig. 7B. Indeed, the PlMAPK2 regulates laccase activity in P. litchii. However, some of them are upregulated and some are downregulated. The authors didn’t go further to illustrate why and how the downregulated or upregulated laccase activity contributes to pathogen plant infection.
L208: I don’t think the summary of “knockout of PlMAPK2 decreased the laccase activity of P. litchi” holds true based on the qRT-PCR results of the eight laccase genes.
Please check the spellings and some sentences throughout the manuscript. Please double-check the citation section. It looks like some citations are not in the correct format (i.e., L319, L370, etc.).
Reviewer 2 Report
Dear authors,
The manuscript Huang et al., disclose the importance of MAPK cascades in zoosporogenesis and pathogenicity in Peronophythora litchii. The aims were clear and methods were well designed to study this signaling pathway.
I've only minor concerns about the manuscript.
1) Regarding Figure 1.B and respective methods aren't clear which are the condition of analysis for asexual stages (0 hpi????). In the infection stage (hip) why do you didn't a negative control? If you did it, you should put it.
Please improve the graph information and clarify the methods.
2) The second concern is related to figure 2. The gel in figure 2.B should be improved. For the quality of this journal IJMS, the gel should be a ladder and negative control.
Best,
Reviewer 3 Report
I have several grammatical changes to add .
AS mentioned in results you should emphasize that the KO results in a slight reduction in virulence.
All of us have seen examples of other genes where KOs clearly impair virulence much more so than you see here.

Reviewer 4 Report
The authors in their work study the involvement of the PlMAPK2 in zoosporogenesis and pathogenicity of P. litcii using two mutant strains generated. They succeed to show that PlMAPK2 affects both functions to a certain extent. The manuscript is well written in general, however, requires a number of corrections and/or clarifications. Following are my specific comments to the authors:
- Line 10. Spelling “functions”.
- Lines 47-52. What is the phylogenetic relation or sequence similarity of the referred kinases to PlMAPK2? If there is no clear similarity/homology the authors should make a comment why do they refer to these kinases (either in the introduction or in the discussion parts). If there is a relation regarding the functions does this connect to sequence similarities? The authors should make a comment on this. Otherwise, there will always be a function to describe for a kinase (by default).
- Lines 55-56. The authors choose ScFus3 and describe it as homologue, although, there is a sequence similarity to only 53 % (line219). This should be stated in my opinion in the introduction part. Is this the biggest % percentage found among fungal/oomycete species regarding homology? The authors should provide a bioinformatic (BLAST?) analysis results regarding homology, and support/argument on why do they choose/refer to a homologue with only 53% similarity. An additional Figure and comments is thus also required before Figure 1A in the results section.
- Lines 64 to 68. Are there any research works regarding the function(s) of these orthologs. Either yes or no, the authors should comment on this (in Results or/and Discussion parts), particularly, if there are reports and how these connect with the present study.
- Line 76. Spelling “qTR-PCR”.
- Line 106. The authors do not show “production” of zoospores but release arrest, thus the legend title should change.
- Line 111. Spelling “Requires”.
- Lines 132-133. The authors refer to “mutants”, although, they show the results for only the M115 mutant in Figure 5. They should amend the figure with the results regarding the other mutant as well (even if there is no significant effect) or they should correct “mutants” to “mutant M115” and describe verbally on what happens in the M113 mutant.
- Lines 133-134. The result does not suggest that the PlMAPK2 may be, or is, associated with the “regulation” of transcription. Regulation is something very specific and there are certain experiments that show this. The authors are encouraged to set their conclusion in a different way (e.g., affects transcription).
- Line 134. The PlMYB1 is reported twice instead of MAD1. Please correct.
- Line 136. Spelling (?) “PlMAPK10” instead of “2”?
- Line 142. Does this experiment study virulence or disease severity? Please clarify and change accordingly if necessary.
- Line 156. Spelling “regulate”.
- Line 188. The authors refer to “new insight”. In order to have a “new insight” there should be reference to the already existing knowledge. There is no comment on that. Please add a few lines regarding this issue or provide an alternative expression.
- Lines 212-213. There is a reference needed here regarding what is stated for PlMAPK10.
- Lines 215-224. There is a loose connection for the Fus3 and PlMAPK2, followed by a somehow not very strong argumentation in this paragraph (see also comment No 3). Please reform the paragraph and give further support to your findings.
Round 2
Reviewer 4 Report
The authors in the revised version of their manuscript replied to all comments but three.
In specific comments No 2, 3 and 6 were not covered.
The authors are encouraged to provide a phylogenetic analysis at the protein level of their target kinase, including other kinases along with these reported in their manuscript, in order to cover comments No2 & No3. The authors could provide a relative figure prior to Figure 1A (as stated in the comments of the first revision round), and provide the relative texts (in results and discussion sections) describing the phylogenetic relation of their protein in respect to homologous and orthologous proteins, putatively supporting relation in functions as well.
Regarding the comment No 6, the observation that the mutant sporangia do not release zoospores does not imply or indicate that there are no zoospores produced, or that they are malformed. The observation just states that there is an effect in their release, in the mutants. The relevant texts and conclusion should be revised in the manuscript accordingly, or additional experiments showing either malformation or no production of zoospores should be provided, in order to support the argument that there is an effect in zoosporogenesis, moreover than this affecting their release.
Round 3
Reviewer 4 Report
The authors covered comment No 6.
The authors have not covered fully comments No 2 and 3, since the phylogenetic analysis provided is not clarifying at all the relationship of PlMAPK2 with other related kinases both in Phytophthora and other Pythiaceae.
In specific, there are only a limited number of MAP Kinases used in the phylogenetic analysis provided by the authors, and these are the so far subjectively selected ones. The authors should use a sufficient number (above 30-35) of relevant protein sequences MAPKs and most relevant Kinases, covering other Phytophthora species (there are more than 160 species cited for Phytophthora), covering all Phytophthora clades, and putatively members of Pythium spp, apart from the sequences already provided. The authors refer to 14 predicted sequences for P. litchii in the introduction. Only these cover half of the proposed number of proteins that could be used in the analysis. Furthermore they should provide (both in M&M and the figure) the relevant sequence IDs with which these are cited in the corresponding databases, in order for the reader to trace the proteins.
Round 4
Reviewer 4 Report
I have no further comments for the authors. The authors covered the last query with the work amended.